# DisTorch: A fast GPU implementation of 3D Hausdorff Distance

**Jérôme Rony** [ID]                                          JEROME@RONY.FR

**Hoel Kervadec** [ID][1]                                    H.T.G.KERVADEC@UVA.NL

[1] *Universiteit van Amsterdam (UvA), Instituut van Informatica (IvI), qurAI group*

**Editors:** Accepted for publication at MIDL 2025

**Keywords:** GPU accelerated, Hausdorff distance, 3D metrics

## 1. Introduction

Distance based metrics are some of the key metrics recommended to assess and analyse the performance of deep learning methods for image semantic segmentation (Maier-Hein et al., 2024). Yet, as authors and reviewers, we find that many works do not include those key metrics. We argue that the following three factors can partly explain the slow adoption of distance based metrics:

- *runtime*: for 3D volumes, common in medical imaging, some implementations easily take minutes *per scan*, translating into hours over a large dataset;
- *complexity of implementation*: faster implementations tend to be more complex, making it difficult to understand and assess their correctness;
- *package management*: implementations comes with their own package requirements and incompatibilities, which can be tedious to integrate into an existing codebase.

While manageable, these factors add friction. In our research context, where time and resources are limited, priorities are chosen which too often involve not picking those metrics. At the same time, the complexity of the implementations partly explain why computed metrics sometimes vary significantly across implementations (Podobnik and Vrtovec, 2024).

In this short paper, we present a simple yet fast implementation of the Hausdorff distance, leveraging the `KeOps` package (Charlier et al., 2021). After briefly describing the core ideas of the implementation, we compare the runtime and memory cost of different implementations, on different 3D volumes from different tasks. Our implementation is freely available under the BSD license: https://github.com/jeromerony/distorch.

## 2. Formulation

### 2.1. Semantic segmentation

A segmentation can be defined as function $s^{(\cdot)} : \Omega \to \mathcal{K}$ that maps each voxel of an image space $\Omega \subset \mathbb{R}^D$, $D \in \mathbb{N}^+$ the number of dimensions, to a set of labels $\mathcal{K}$. From this segmentation (often implemented as a dense array/tensor), we can define different closed subsets, representing the areas predicted as class $k$: $\Omega_s^{(k)} := \{p \in \Omega | s^{(p)} = k\}$. It follows that all those subsets are exactly disjoint, *i.e.,* $\Omega_s^{(k)} \cap \Omega_s^{(l)} = \emptyset \quad \forall k \neq l$, and that they union covers the whole image space: $\cup \{\Omega_s^{(k)}\}_{k \in \mathcal{K}} = \Omega$. Their boundaries are part of their respective subsets: $\partial \Omega_s^{(k)} \subseteq \Omega_s^{(k)}$.

## 2.2. Boundary and Hausdorff distance

The Hausdorff distance is defined between two segmentation (denoted here as $s$ and $g$), and represents *"the maximum distance of the minimum distance between each pair of elements"*, computed over their boundaries:

$$\mathrm{HD}(s,g;k) := \max \left\{ \max_{i \in \partial\Omega_s^{(k)}} \min_{j \in \partial\Omega_g^{(k)}} d(i,j), \max_{i \in \partial\Omega_g^{(k)}} \min_{j \in \partial\Omega_s^{(k)}} d(i,j) \right\} \tag{1}$$

with $d(a,b) \in \mathbb{R}_+$ representing the Euclidean distance between the two coordinates. Computing the Hausdorff distance (and associated metrics) can be done in four steps:

1. extract the two boundaries $\partial\Omega_s^{(k)}$ and $\partial\Omega_g^{(k)}$;
2. compute the *Euclidean Distance Transform* (EDT) of the two boundaries;
3. perform a masking of this EDT with the boundary masks;
4. pick the maximum, 95-th percentile or average of those masked values, depending on the metric computed.

Computing the EDT twice is usually the costly operation of this metric, often CPU bound[1].

## 2.3. Our approach

In contrast, we do not compute the EDT. Instead, we observe that $|\partial\Omega_s^{(k)}| \ll |\Omega_s^{(k)}|$, which in a way has one less dimension compared to the full volume (as we consider only its surface). For instance, a blob-like shape in a 2D image has a contour of *very* approximate length $2(h+w)$, where $h$ and $w$ are the height and width of the image. Henceforth, we directly compute each pairwise distances between $\partial\Omega_s^{(k)}$ and $\partial\Omega_g^{(k)}$, which has a quadratic complexity.

From a distance, it appears to be a bad idea. But with modern programming frameworks leveraging the parallelism capabilities of modern hardware (Charlier et al., 2021), this turns out to be quite fast. Moreover, we do not compute quite all pairwise distances between $\partial\Omega_s^{(k)}$ and $\partial\Omega_g^{(k)}$: as some elements are common to both boundaries they can be skipped (their distances being 0). As such, we compute the following quantity:

$$\begin{aligned}
\mathrm{HD}(s,g;k) = \max\{ &\max_i \min_j \{d(i,j) \,|\, i \in (\partial\Omega_s^{(k)} \setminus \partial\Omega_g^{(k)}), j \in \partial\Omega_g^{(k)}\}, \\
&\max_i \min_j \{d(i,j) \,|\, i \in (\partial\Omega_g^{(k)} \setminus \partial\Omega_s^{(k)}), j \in \partial\Omega_s^{(k)}\}\}.
\end{aligned} \tag{2}$$

Performing this computation still requires computing a pairwise distance matrix (akin to a kernel matrix) between large sets, which is expensive, and will often result in Out-Of-Memory errors if done naively. However, the `KeOps` library (Charlier et al., 2021) was designed for that specific purpose: it allows computing reductions (*e.g.* sum, max, min, etc.) over kernel matrices efficiently, by never materializing the full matrix in memory. This avoids unnecessarily filling the memory with unused intermediate results, and remains quite fast by taking advantage of faster memory locations (*i.e.* GPU registers). Additional care needs to be taken for edge cases such as empty boundary sets and matching boundary sets.

---

1. https://docs.scipy.org/doc/scipy/reference/generated/scipy.ndimage.distance_transform_edt.html is reused in many implementations.

Table 1: Comparison of the runtime in ms (per sample) and GPU memory usage in GiB (maximum) of different implementations. NA stands for Not Applicable.

|  | Segthor | | OAI | | WMH 1.0 | |
|---|---|---|---|---|---|---|
|  | Runtime | Mem. | Runtime | Mem. | Runtime | Mem. |
| MedPy | $2.6{\times}10^4$ | NA | $1.8{\times}10^4$ | NA | 296 | NA |
| MeshMetrics | $8.5{\times}10^3$ | NA | $1.2{\times}10^4$ | NA | 436 | NA |
| Monai | 723 | 4.7 | $1.7{\times}10^3$ | 2.1 | 52.2 | 0.52 |
| Monai w/ `cuCIM` | 24.9 | 2.6 | 22.4 | 0.95 | 6.3 | 0.09 |
| Ours | 29.1 | 1.7 | 26.8 | 0.62 | 1.4 | 0.06 |

## 3. Experiments

We benchmark our implementation on three datasets: SegTHOR (Lambert et al., 2020) (CT of heart, esophagus, aorta and trachea, with a size of $512{\times}512{\times}{\sim}200$), OAI (Almajalid et al., 2022) (CT of knee bones and cartilages, $384{\times}384{\times}160$), and WMH 1.0 (Kuijf et al., 2022) (multi-sequence MRI of White Matter Hyperintensity, ${\sim}128{\times}{\sim}256{\times}\sim 80$). The computation were done between the reference labels and some network prediction (results of a sister-submission of this paper). Not only varying in size of the 3D volumes, the size of the objects varies across classes and tasks, covering a variety of big, small, thin, thick, compact and less compact objects.

Benchmark results are reported in Table 1, against MedPy, MeshMetrics (Podobnik and Vrtovec, 2024) and Monai (Cardoso et al., 2022) with(out) `cuCIM` optional dependency. Note that MedPy and MeshMetrics are CPU-only implementation. Everything was run on the same machine with an AMD 5800X CPU, 32GB of DDR4 RAM, and a NVIDIA RTX 4090. Due to space limitation, we will only mention that we ensured computed metrics were consistent with Monai. We used one Python environment per implementation, to sidestep packages incompatibilities (notably, Monai is currently incompatible with `NumPy` 2.0, and `cuCIM` is not compatible with Python 3.13 yet). We believe it does not invalidate the benchmark but rather illustrates how cumbersome it can be to integrate extra dependencies.

## 4. Discussion & conclusion

By leveraging `KeOps`, our implementation remains quite simple (the core of its implementation being less than 50 lines of code) and provides a comparable runtime to Monai when it relies on the optimized `cuCIM` library. Those two are consistently much faster than alternatives, by a few orders of magnitude. Zooming-in, our implementation is even faster than Monai when the 3D volumes only contain small objects (*e.g.* with WMH 1.0), while having a lower memory footprint overall (which could prove useful for very big volumes). The converse will also be true: our implementation's performance will suffer from random-looking predictions (possibly early in training) or manually crafted worst-case scenarios. We argue that in such cases, computing the HD distance does not make much sense anyway, and should be done only when other metrics (*e.g.* Dice) are sufficiently high.

We believe that this simple implementation strikes a good balance between ease of understanding, use, and speed, which makes it a good candidate to integrate in existing codebases without significantly refactoring them.

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
