# OpenReview forum: "DisTorch: A fast GPU implementation of  3D Hausdorff Distance"
_MIDL.io/2025/Short_Papers — MIDL 2025 - Short Papers_

### Official Review · Reviewer_TBo6 · 2025-04-25

**Rating:** 3
**Confidence:** 4

**Summary:**

The manuscript presents a different implementation of the Hausdorff Distance that accelerates its computation.

**Strengths:**

+ Relatively high reductions in runtime

**Weaknesses:**

- It is not really clear to me that without specifci context this runtime reduction is of particularly importance; of course, it is nice, but the runtime of other methods is still on the order of seconds; anyway, this is a more minor comment perhaps mostly alluding to the framing of the manuscript.

- There is no demonstration that the implmentation also calculates the same Hausdorff distance. This seems particularly important as one of the reference explicitly states that different implementations result in different outputs. How can it be asserted that this implementation calculates the "correct" metric?

---

### Decision · Program_Chairs · 2025-05-01

Accept